# Health Anxiety Predicts Postponing or Cancelling Routine Medical Health Care Appointments among Women in Perinatal Stage during the Covid-19 Lockdown

**DOI:** 10.3390/ijerph17218272

**Published:** 2020-11-09

**Authors:** Mehran Shayganfard, Fateme Mahdavi, Mohammad Haghighi, Dena Sadeghi Bahmani, Serge Brand

**Affiliations:** 1Department of Psychiatry, Arak University of Medical Sciences, Arak 3848176341, Iran; mshayganfard@arakmu.ac.ir; 2Endocrinology and Metabolism Research Center, Arak University of Medical Sciences, Arak 3848176341, Iran; 3Student Research Committee, Arak University of Medical Sciences, Arak 3848176341, Iran; fatememahdavi111@gmail.com; 4Research Center for Behavioral Disorders and Substances Abuse, Hamadan University of Medical Sciences, Hamadan 65174, Iran; dr_haghighi_ps@yahoo.com; 5Departments of Physical Therapy, University of Alabama at Birmingham, Birmingham, AL 35209, USA; dena.sadeghibahmani@upk.ch; 6Sleep Disorders Research Center, Kermanshah University of Medical Sciences, Kermanshah 67146, Iran; 7Center for Affective-, Stress- and Sleep Disorders (ZASS), University of Basel, Psychiatric Clinics (UPK), 4002 Basel, Switzerland; 8Division of Sport Science and Psychosocial Health, Department of Sport, Exercise, and Health, University of Basel, 4052 Basel, Switzerland; 9Substance Abuse Prevention Research Center, Health, Institute, Kermanshah University of Medical Sciences, Kermanshah 67146, Iran; 10School of Medicine, Tehran University of Medical Sciences, Tehran 25529, Iran

**Keywords:** COVID-19, perinatal care, health anxiety, depression, stress, pregnancy

## Abstract

To avoid spreading the Corona Virus Disease 2019 (COVID-19), health authorities have forced people to reorganize their working and private lives and to avoid open and public spaces as much as possible. This has also been the case for women both during pregnancy and after delivery. Here, we investigated the associations between subjective beliefs in risk of infections and health anxiety, depression, stress, and other perinatal dimensions. To this end, we assessed 103 women (mean age: 28.57 years) during pregnancy and after delivery. They completed a series of questionnaires covering sociodemographic information, perinatal information, health anxiety, post-partum depression, and stress. Sixty-six participants (64.1%) were in the pre-partum stage, and 37 (35.9%) were post-partum. Health anxiety was unrelated to depression or stress. Knowing and being close to infected people was associated with higher health anxiety. Strict following of the safety recommendations was associated with greater health anxiety, depression, and stress. Postponing or cancelling routine medical check appointments was observed among participants with high health anxiety scores. Higher illness severity, overall health anxiety scores, and lower stress scores predicted those participants who postponed or cancelled their routine medical check appointments. Post-partum stage and a larger number of children were associated with higher stress scores, but not with depression or stress. The results are of practical and clinical importance; it appears that health anxiety, which is to say fear of getting infected with COVID-19 during pregnancy or at the post-partum stage, was associated with postponing or cancelling routine medical check appointments, but not with stress or depression.

## 1. Introduction

In Iran, the emergence of the Coronavirus Disease 2019 (COVID-19) and the associated epidemic risk forced health authorities to protect their citizens from both getting infected and spreading the virus to others [1,2]. To this end, restrictions were placed on going into open and crowded spaces; borders, educational institutions (kindergartens, schools, universities), cultural and sports areas, and religious and sacred places were (temporarily) closed. As of the end of April 2020, of 330,137 tested patients, 80,868 had been infected with COVID-19. Of these, 55,987 had recovered, 3513 remained critically ill and 5031 people had died [3]. Announcement of the lockdown also appears to have caused panic and anxiety [1,2].

Typically, epidemic danger is associated with unpredictability and uncertainty [4]. In this particular case, unpredictability and uncertainty are associated with anxiety about getting infected by COVID-19. Among 1789 Iranian adults responding to an online survey on this topic, 68% reported moderate to severe anxiety about getting infected. Women and participants with a modest health status had 1.91- to 3.46-fold increased odds of recording moderate to high anxiety scores [5]. Higher scores for self-reported levels of stress, anxiety, and depression have been reported not only in the general population [6] but also within and between different countries [7].

Inspecting the evidence in more detail reveals that levels of anxiety were higher among women than men, among people following COVID-19 news, among adults aged 20 to 40 years, and among individuals with at least one family member, relative, or friend who had contracted the disease [8]. Compared to those who survived an infection, individuals who later died from the disease had higher self-reported anxiety and depression scores and higher cortisol concentrations. Higher anxiety scores and higher serum cortisol levels were associated [9].

As regards women in the perinatal stage of a pregnancy, there is an argument that morning sickness emerged during evolution to protect the fetus and embryo from toxins [10]. It would therefore seem that, compared to men and women who are not pregnant or have not recently given birth, women during the perinatal period need greater protection against physiological and psychological harm, as well as increased health care and attention. This increased attention is required for the unborn or newly born child as well as for the benefit of the women’s physical and psychological state. Psychological distress during the perinatal period can be associated with psychophysiological changes increasing the risk of perinatal/postnatal symptoms of depression [11]. Relatedly, among pregnant women, the prevalence of a clinical anxiety disorder was found in one study to be 15.2%, while the prevalence of a generalized anxiety disorder was 4.1% [12]. Importantly, higher anxiety scores during pregnancy and the perinatal period might negatively influence outcomes for the child [12,13]. Results from a meta-analysis indicate that psychological, social, and physiological factors may increase the risk of both new onset anxiety and a worsening of anxiety during the perinatal period [14]. Results from systematic reviews have also shown the dimensions of anxiety, depression, and stress during the perinatal period to be interrelated [15,16]. From an evolutionary point of view, anxiety during the perinatal period might help pregnant women avoid exposure to toxins and risky behavior [17,18,19]. This mechanism may help to explain why during the COVID-19 pandemic pregnant women have recorded higher scores for anxiety, stress, and depression [20,21,22,23,24], and have displayed avoidance behavior with respect to hospital visits [21,25], gynecologic triage and ultra-sound units [26], and more safety behaviors such as opting for epidural analgesia [26], cesarean section, bottle feeding, and postnatal rest at home [25].

The cognitive-emotional concept of health anxiety is similar to anxiety, but closer to a subjective uncertainty about staying or remaining healthy. Health anxiety is understood as a dimensional construct ranging from “not observable” to “extremely severe” [27,28]. The most intensive form of health anxiety is hypochondriasis or somatic symptom disorder and illness anxiety disorder following the DSM-5 [29]. In the present study, we decided to assess health anxiety and not state or trait anxiety; unlike state or trait anxiety, health anxiety concerns the cognitive-emotional belief that one’s health is in danger. As such, health anxiety is considered a health belief model. Studies on health belief models [30,31,32] have shown that subjective beliefs impact on a person’s susceptibility to illnesses. Thus, healthy belief models, and in this case health anxiety, influence the degree to which a woman in the perinatal stage believes she is more or less susceptible to infection by the COVID-19 disease.

Given this background, in the present study we combined psychological findings related to the COVID-19 pandemic with the particular status of females during the perinatal period. More specifically, we investigated whether health anxiety, stress or depression led to postponement or even cancellation of routine medical health care appointments, given that such forms of avoidance behavior might increase the risks of additional health issues both for the (future) mothers and for their children.

The following three hypotheses and three research questions were formulated. First, following others [6,15,16], we expected that self-rated dimensions of health anxiety, depression, and stress would be related. Second, following Moghanibashi-Mansourieh [8], we hypothesized that being in touch with an intimate (family member, close friend, neighbor) testing positive for COVID-19 would increase the odds of higher health anxiety scores. Third, following both the evolutionary hypothesis of health anxiety as a protective strategy to avoid contact with toxins and to avoid risky behavior [17,18,19] and recent findings indicating avoidance behavior by pregnant women during the pandemic [21,25,26], we anticipated that higher health anxiety scores would be associated with higher odds of postponing or cancelling routine health care appointments. We treated as exploratory the question of which dimensions (health anxiety, depression, stress, age) would predict those participants postponing or cancelling their routine medical health care appointments. The second exploratory research question concerned the relation of health anxiety, stress and depression to sociodemographic dimensions such as age, number of pregnancies (including prematurely terminated pregnancies), number of children, employment status, and level of education. The third exploratory question was whether participants recorded different scores on health anxiety, depression, and stress before versus after delivery.

To test the hypotheses and to address the research questions, women in the perinatal period completed a series of questionnaires. We believe that this study has the potential to identify those women during perinatal stage who are at particular risk of postponing or cancelling routine health care appointments.

## 2. Method

### 2.1. Study Procedure

Women in the perinatal period and registered as patients with the Department of Gynecology and Midwifery of the Arak University of Medical Sciences (ARAKMU; Arak, Iran) were approached to participate in the present cross-sectional study on the associations between symptoms of depression, stress, and anxiety during the Covid-19 lockdown. All eligible participants were informed about the aims of the study and the secure and anonymous data handling. Thereafter, participants signed a written informed consent. They then completed a booklet of questionnaires covering sociodemographic information, information regarding pregnancy, delivery and the post-partum stage, health anxiety (Health Anxiety Inventory; see the description of the questionnaires below), post-partum depression (Edinburgh Postnatal Depression Scale; EPDS; see below), perceived stress (Perceived Stress Scale; see below), and specific COVID-19-related questions (see below).

The ethical committee of the Arak University of Medical Sciences (ARAKMU; Arak, Iran) approved the study (code: IR.ARAKMU.REC.1399.016), which was performed in accordance with the seventh and current version [33] of the Declaration of Helsinki.

### 2.2. Participants

Of 120 women approached, 103 (85.84%) agreed to participate. Inclusion criteria were: (1) At least 18 years old; (2) either in a prenatal stage or postnatal stage no longer than six weeks after delivery; (3) Willing and able to comply with the study requirements. (4) Signed written informed consent. Exclusion criteria: (1) Current or past (six months before pregnancy) severe psychiatric issues such as major depressive disorders, bipolar disorders, schizophrenia spectrum disorder, severe sleep disorders, substance use disorder; to this end an experienced psychiatrist or clinical psychologist performed a brief psychiatric interview based on the DSM-5 [34]. (2) Current severe physiological issues such as diabetes or allergies, including surgery with narcosis, which by definition temporarily alters psychophysiological status.

Being infected with COVID-19 was not an exclusion criterion though at the time of assessment all COVID-19 tests were negative (RT-PCR; Roche-Cobas-SARS-Cov-2-Target1/Target2; Cobas6800^®^; Roche, Basel, Switzerland).

### 2.3. Tools

#### 2.3.1. Sociodemographic Information

Participants reported their age (in years), highest educational level (compulsory school; diploma; higher diploma; higher educational training), employment status (employed: yes vs. no), number of children, and number of pregnancies.

#### 2.3.2. Health Anxiety

To assess health anxiety, participants completed the Persian version [35] of the Health Anxiety Inventory [27]. The questionnaire consists of 18 questions focusing on anxiety about one own’s health, ranging from no fear of health at all to hypochondriasis (DSM-IV; [36]) or somatic symptom disorder and illness anxiety disorder (DSM-5; [29]). Hypochondriasis is understood as the clinical and dysfunctional belief that one is suffering from or going to be affected by a severe and dangerous disease. Typical items are: 1. (a) “I do not worry about my health”; (b) “I occasionally worry about my health”; (c) “I spend much of my time worrying about my health”; (d) “I spend most of my time worrying about my health”; 2. (a) ”I notice aches/pains less than most other people (of my age)”; (b) “I notice aches/pains as much as most other people (of my age)”; (c) “I notice aches/pains more than most other people (of my age)”; (d) “I am aware of aches/pains in my body all the time”. 3. (a) “As a rule, I am not aware of bodily sensations or changes”; (b) “Sometimes I am aware of bodily sensations or changes”; (c) “I am often aware of bodily sensations or changes”; (d) “I am constantly aware of bodily sensations or changes”. Items are aggregated to the following dimensions: Illness severity, illness likelihood, body vigilance, and the total score. Higher sum scores reflect a more pronounced severity of health anxiety.

#### 2.3.3. Behavior Related to the COVID-19 Pandemic

Three self-administered items focused on COVID-19-related behavior:

1.“Do you live with a person (family member; relative; colleague at work) currently infected with COVID-19?” Response alternatives, treated as a Likert scale, were the following: No (=1); yes, but not no direct contact (=2); yes, and with direct contact (=3).2.“How necessary is it to adhere to the hygiene and social distancing recommendations?” Response alternatives, again treated as a Likert scale, were: Not necessary (=1); it is necessary, but I do not strictly follow the rules (=2); it is necessary, and I do follow the rules (=3); this is absolutely necessary; when I see that other people do not follow the rules, I get worried and feel uncomfortable (=4)3.“Have you postponed or cancelled routine medical health care appointments for your and/or your child?” Response alternatives were: No (=1); yes (=2).

#### 2.3.4. Self-Ratings of Symptoms of Depression

Participants completed the Persian version [37] of the Edinburgh postnatal depression scale EPDS [38], which consists of ten items. Anchor points for each item are 0 and 3. Higher total scores reflect more marked symptoms of postnatal depression. The maximum score is 30. (Cronbach’s alpha = 0.83).

#### 2.3.5. Perceived Stress

Participants completed the 14-items Perceived Stress Scale (PSS [39]), a self-report measure designed to assess whether and to what extent situations in a person’s life are considered as stressful, unpredictable, uncontrollable, and overloading. The Persian version has been psychometrically validated [40,41,42]. Typical items are: “In the last month how often have you felt that you were unable to control the important things in your life”, or “…how often have you been upset because of something that happened unexpectedly?”, or “… how often have you felt confident about your ability to handle your personal problems?”. Answers are given on 5-point rating scales ranging from 0 (=never) to 4 (very often). Some questions are positively worded and some are reverse coded. Higher sum scores reflect a higher perceived stress (Cronbach’s alpha = 0.83).

### 2.4. Statistical Analysis

With a series of Pearson’s correlations associations between participants’ age, health anxiety, depression, and stress were calculated. Three multivariate ANOVAs were performed with the three COVID-19 related questions as independent factors, and health anxiety, depression, and stress as dependent variables. For post-hoc analyses, Bonferroni-Holm corrections for p-values were used. A binary logistic regression was performed to predict postponing/cancelling medical routine health care appointments (yes vs. no) as a function of health anxiety, depression, stress, and age. A series of t-tests and X^2^-tests was performed to investigate differences in health anxiety, depression, stress, and COVID-19-related issues between participants before and after delivery. The nominal level of significance was set at alpha <0.05. Statistical computations were performed with SPSS^®^ 26.0 (IBM Corporation, Armonk, NY, USA) for Apple Mac^®^.

## 3. Results

### 3.1. General Overview

A total of 103 women (mean age: 28.57 years; SD = 6.85) took part in the study. Of these, 66 (64.1%) were assessed during pregnancy (second and third trimester), and 37 (35.9%) were assessed after delivery. Table 1 provides the descriptive and inferential statistical indices of sociodemographic information, scores of health anxiety, depression, and stress, and COVID-19-related issues, both for the entire study sample, and separately for participants before and after delivery. Statistical indices reported in tables are not repeated in the text.

Compared to pregnant participants, participants after delivery reported significantly higher stress and a greater number of children. For all other dimensions (age, number of pregnancies, health anxiety, depression, COVID-19-related questions, education and employment status), no descriptive or statistical differences were observed.

### 3.2. Associations between Age, the Number of Children, Health Anxiety, Depression, and Stress

Table 2 provides the correlations for age, number of children, health anxiety, depression, and stress.

Correlations between age, health anxiety, depression, and stress were trivial to low. A greater number of children was associated with higher stress and with older age.

### 3.3. Concerns about COVID-19 and Symptoms of Health Anxiety, Depression, and Stress

Table 3 provides the descriptive and inferential statistical indices for health anxiety, depression, and stress as dependent variables and the three COVID-19-related questions as independent factors.

Participants living with or being in contact with a person with COVID-19 had significantly higher health anxiety scores. No differences were found for depression or stress.

Participants adhering strictly to the rules and feeling worried and uncomfortable recorded significantly higher health anxiety and depression scores. No differences were found for stress.

Participants postponing/cancelling routine medical care appointments had significantly higher health anxiety and depression scores. No differences were found for stress. Figure 1 displays the health anxiety scores of participants who respectively did and did not postpone or cancel routine medical care appointments.

### 3.4. Postponing/Cancelling Routine Medical Care Appointments

To predict who did or did not postpone/cancel routine medical care appointments, a binary logistic regression was performed (see Table 4 for details). The accuracy of the model in assigning participants to the respective dichotomous groups (postponing/cancelling: yes vs. no) was 92.2% (54 out of 57 participants correctly assigned as “yes”; 41 out of 46 correctly assigned as “no”). Predictors explained 85% of the variance of the outcome variable (Nagelkerke R^2^: 0.85). Higher illness severity scores and higher total health anxiety, but lower stress predicted which participants postponed/cancelled routine medical care appointments. Illness likelihood, body vigilance, and depression were excluded from the equation as their indices did not reach statistical significance.

## 4. Discussion

The key results of the present study were that among women in a perinatal stage health anxiety was unrelated to depression or stress. However, health anxiety was related to being close to people infected with COVID-19, and to worrying about getting infected with COVID-19. Furthermore, higher illness severity and overall health anxiety, and lower stress scores predicted those participants postponing or cancelling routine medical health care appointments. Next, women in the postpartum stage and having more children reported higher stress. The present results add to the current literature in the following important ways. First, health anxiety was not related to depression or stress but was related to the risk of getting infected by intimate associates with COVID-19. Second, on a behavioral level, health anxiety was related to postponing or cancelling routine medical health care appointments (both for the mothers and their children); avoidance behavior of this kind carries the risk that health issues will be missed or overlooked.

Three hypotheses and three exploratory research questions were formulated, and each of these is considered in turn.

Our first hypothesis was that scores for health anxiety, depression, and stress would be associated, but this was not supported; the correlations between health anxiety, depression, and stress were trivial to low. Higher stress was associated with a greater number of children which in turn was related to participants’ older age. Given this, the present results to not confirm previous findings [6,15,16,20,21,22,24,25,43].

Our second hypothesis was that women in the perinatal stage and living close to people infected with COVID-19 would record higher health anxiety scores, and this was supported. In this respect the present results are consistent with findings reported in a previous study [8]. However, we believe that the present data do add to the current literature in an important way: Health anxiety was unrelated to depression or stress, but to the subjective sense of avoiding possible exposure to the virus.

The evidence available to us from this study is unable to illuminate the cognitive-emotional mechanisms that might account for the results relating to these two hypotheses. Given this, we advance the following possibilities. First, while previous studies have assessed (state or trait) anxiety, we assessed health anxiety, which is a person’s (dysfunctional) cognitive-emotional health belief model. As such, a health belief model will not necessarily be related to objective stressors or to symptoms of depression but to the dysfunctional belief that one’s body might become severely ill, irrespective of any objective danger. Second, and relatedly, even if a woman in the perinatal stage is living close to a person infected by COVID-19, precautions such as social distancing and scrupulous respect of hygiene rules could objectively and securely have protected her from getting infected. However, it is characteristic of health anxiety and health beliefs models that they do not take objective circumstances into consideration [30,31,32].

Our third hypothesis was that health anxiety would be related to postponing or cancelling routine medical health care appointments and this was supported. Given that no research question stated in this specific form has previously been addressed, we refer to the concepts of evolutionary psychology and evolutionary psychiatry. Health anxiety is considered a protective strategy to avoid contact with toxins and to avoid risky behavior [17,18,19]. Thus, from an evolutionary point of view, avoidance of danger makes sense. However, and in the present specific context, avoidance in terms of postponing or cancelling routine medical health care appointments can put mothers’ and children’s health at risk in the longer term, given that the key aim of such routine medical health care appointments is to detect and address potential health issues. As such, women in the perinatal stage with high health anxiety appear to create a conflict between short-term benefits (harm avoidance) and long-term costs (preventing and dealing with health issues). The following observations give some support to the mechanisms described above. During the COVID-19 pandemic, 25.8% of 2740 pregnant Canadian women stopped their in-person visits, and the main reason given was concern about getting infected [21]. During the COVID-19 pandemic, pregnant women in Wuhan (China) often refused to go to any hospital, and 12.7% more women than planned refused vaginal delivery and opted for cesarean section [25]. Visits to obstetric triage, gynecological triage, and ultra-sound units decreased by one third in an Israeli hospital; additionally, more epidural analgesia was requested and nulliparous asked more often for operative vaginal deliveries [26]. A higher rate of pregnant women in Sardinia asked for voluntary termination of pregnancy; others asked for invasive prenatal screening [43].

The first exploratory research question concerned the degree to which health anxiety, depression, stress, and age might independently or in combination predict postponement or cancellation of routine medical health care appointments. It turned out that higher health anxiety traits and *lower* stress predicted those participants reporting these kinds of avoidance behavior, thus confirming what has previously been observed and discussed. However, it was unexpected that lower stress was associated with higher avoidance given that stress and health anxiety were unrelated. The quality of the data does not allow a deeper understanding of this finding. However, we believe that various confounding factors were involved. Higher stress was associated with older age and with a greater number of children; complementary to this it is possible that younger participants with fewer children were more prone to avoid routine medical health care appointments. Thus, though it is admittedly highly speculatively, it is possible that the older participants with more children and more experience to draw upon made more informed decisions about routine medical health care appointments.

Our second exploratory research question addressed the possibility that sociodemographic and perinatal dimensions are related to dimensions of health anxiety, depression, and stress, and the answer was no.

The third research study question concerned the possibility that participants before and after delivery respectively would have different levels of health anxiety, depression stress, and other perinatal and COVID-19-related issues. As shown in Table 1, participants after delivery recorded higher stress scores. In our opinion this is unsurprising given that caring for an infant is likely to involve a degree of stress.

Despite the novelty of the results, the following limitations warrant against their overgeneralization. First, participants were recruited from only one study center and only participants willing and able to comply with the study conditions took part in the study; it follows that sample-related biases cannot be fully ruled out. Second, the cross-sectional character of the study design precludes conclusions about causal relationships between the variables. Third, a follow-up design would have allowed us to assess, whether postponing or cancelling routine medical health care appointments had consequences for participants’ or their children’s health. Likewise, fourth, it would have been interesting to explore if such avoidance behavior was associated with a lower risk of getting infected with COVID-19. Similarly, fifth, if any participants had become infected by COVID-19, it would have been interesting to calculate whether this outcome could have been predicted from the present data. Sixth, it is conceivable that unassessed and latent variables such as abortion/miscarriage biased two or more dimensions in the same or opposite directions. Seventh, the sample size was small, though we focused on effect size calculations which are unrelated to sample size. Eighth, apart from avoidance behavior, we did not ask if participants used alternative strategies such as online consultation [21,25,44] or phone visits [21]. Furthermore, it turns out that during the COVID-19 pandemic pregnant women with higher physical activity indices have also reported lower anxiety and depression scores [24]. As such, in addition to social support, regular physical activity should be taken into consideration in efforts to improve the physical and mental health of perinatal stage women [24,45].

## 5. Conclusions

Among a sample of women in the perinatal stage and assessed during the COVID-19 pandemic, health anxiety was unrelated to symptoms of depression and stress. Rather, health anxiety was related to proximity to closer others suffering from COVID-19. Importantly, higher health anxiety was related to postponing or cancelling routine medical health care appointments for the participant or her child. Consequently, higher health anxiety could be associated with a greater risk of not detecting or prevent potential health problems. We speculate that women in the perinatal stage with high health anxiety appeared to have a conflict between short-term benefits (harm avoidance) and longer-term disadvantages (preventing and detecting health issues).

## Figures and Tables

**Figure 1 ijerph-17-08272-f001:**
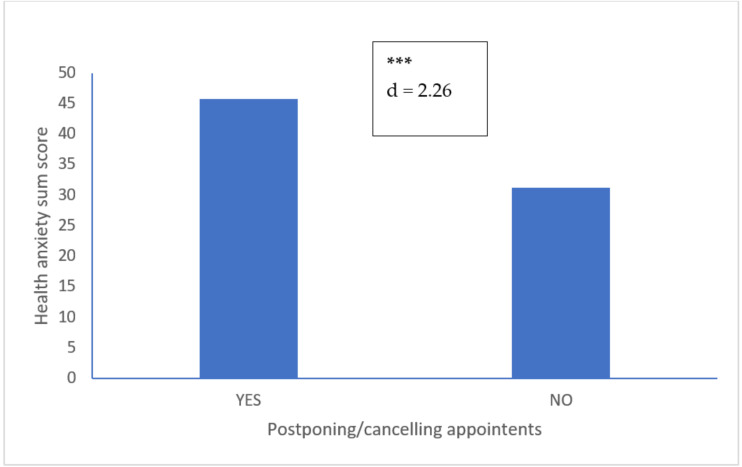
Health anxiety scores of participants who did (*n* = 57) and who did not (*n* = 46) postpone or cancel routine medical appointments. Notes: *** = *p* < 0.001; d = Cohen’s d: Large effect size.

**Table 1 ijerph-17-08272-t001:** Descriptive and inferential statistical overview of sociodemographic, pregnancy-related, and psychological dimensions; the entire sample (*n* = 103) and separately for participants before (*n* = 66) and after delivery (*n* = 37).

		Samples		Statistics	
	Total Sample	Before Delivery	After Delivery	F (1, 101)	η_p_^2^
*n*	103	66	37		
	M (SD)	M (SD)	M (SD)		
Age	28.57 (6.85)	28.97 (7.45)	27.86 (5.62)	0.62	0.006 (T)
Number of pregnancies	2.20 (1.10)	2.35 (1.12)	1.95 (1.03)	3.26	0.031 (S)
Number of children	1.41 (0.99)	1.23 (0.99)	1.73 (0.93)	6.37 *	0.060 (M)
Gestational age (weeks)		27.20 (5.77)	-		
Time lapse after delivery (weeks)		-	4.82 (0.67)		
Health anxiety					
Illness likelihood	24.63 (5.58)	24.85 (5.23)	24.24 (6.23)	0.28	0.003 (T)
Illness severity	5.27 (3.50	5.33 (3.21)	5.16 (4.00	0.06	0.001 (T)
Body vigilance	9.20 (2.34)	9.21 (2.25)	9.19 (2.52)	0.02	0.000 (T)
Total score	29.31 (9.60)	39.70 (8.57)	38.62 (11.24)	0.30	0.010 (T)
Edinburgh Postnatal Depression	12.77 (3.66)	12.50 (4.04)	13.24 (2.84)	1.00	0.010 (T)
Perceived stress	22.23 (7.30)	20.29 (7.46)	25.70 (5.58)	14.81 ***	0.128 (M)
	*n* (%)	*n* (%)	*n* (%)		
Q1; Close person with COVID-19? (no/yes + no contact; yes + contact	61/40/2	39/26/1	22/14/1	X^2^ (*n* = 103; df = 2) = 0.19	
Q2; necessary to adhere to the rules? (yes + I adhere sometimes; yes + always; yes + worries)	10/59/34	8/40/18	2/19/16	X^2^ (*n* = 103; df = 2) = 3.29	
Q3; postponing/cancelling appointments (yes/no)	57/46	39/27	18/19	X^2^ (*n* = 103; df = 2) = 1.05	
	*n* (%)	*n* (%)	*n* (%)		
Education (compulsory school/diploma/higher education)	33/47/23	25/31/10	8/16/13	X^2^ (*n* = 103; df = 2) = 6.36	
Employment (yes, no)	26/77	14/52	12/25	X^2^ (*n* = 103; df = 1) = 1.58	

Notes: * = *p* < 0.05; *** = *p* < 0.001. T = trivial effect size; S = small effect size; M = medium effect size.

**Table 2 ijerph-17-08272-t002:** Overview of Pearson’s correlation coefficients between age, number of children, health anxiety, depression, and stress.

		Dimensions			
	Age	Number Children	Health Anxiety	Depression	Stress
Age	-	0.44 ***	0.16	0.04	0.15
Number of children		-	0.15	−0.00	0.27 **
Health anxiety inventory			-	0.07	−0.04
Edinburgh Depression Scale				-	0.14
Perceived stress scale					-

Notes: ** = *p* < 0.01; *** = *p* < 0.001.

**Table 3 ijerph-17-08272-t003:** Health anxiety, depression, and stress, depending from COVID-19-related questions.

Dimensions												
	Health Anxiety			Statistics	Depression			Statistics	Stress			Statistics
Q1; Close person with COVID-19?	No	Yes, no contact	Yes, with contact		No	Yes, no contact	Yes, with contact		No	Yes, no contact	Yes, with contact	
N	61	40	2	F (2, 100) η_p_^2^	61	40	2	F (2, 100) η_p_^2^	61	40	2	F (2, 100) η_p_^2^
	M (SD)	M (SD)	M (SD)		M (SD)	M (SD)	M (SD)		M (SD)	M (SD)	M (SD)	
	36.62 (8.75)	42.83 (9.52)	51.00 (2.83)	7.42 ***, 0.129 (M)	12.70 (4.19)	12.90 (2.83)	12.00 (1.41)	0.08, 0.002 (T)	23.00 (7.27)	19.50 (2.12)	22.23 (7.30)	0.45, 0.009 (T)
Q2 Necessary to adhere to recommendations?	Yes, sometimes	Yes, always	Yes +worries		Yes, sometimes	Yes, always	Yes +worries		Yes, sometimes	Yes, always	Yes +worries	
N	10	59	34	F (2, 100) η_p_^2^	10	59	34	F (2, 100) η_p_^2^	10	59	34	F (2, 100) η_p_^2^
	M (SD)	M (SD)	M (SD)		M (SD)	M (SD)	M (SD)		M (SD)	M (SD)	M (SD)	
	28.90 (13.61)	36.64 (7.69)	47.00 (4.72)	30.11 ***, 0.376 (L)	11.20 (3.29)	12.31 (4.02)	14.03 (2.69)	3.58 *, 0.067 (M)	21.30 (6.34)	20.85 (7.98)	24.91 (5.54)	3.61 *, 0.067 (M)
	M (SD)	M (SD)			M (SD)	M (SD)			M (SD)	M (SD)	M (SD)	
Q3; postponing/cancelling appointments (yes/no)	Yes	No			Yes	No			Yes	No		
N	57	46		F (1, 101)	57	46		F (1, 101)	57	46		F(1, 101)
	45.82 (4.15)	31.24 (8.13)		139.08 ***, 0.579 (L)	12.54 (3.55)	13.04 (3.82)		0.47, 0.005 (T)	21.14 (7.53)	23.59 (6.85)		2.91, 0.028 (S)

Notes: * = *p* < 0.05; *** = *p* < 0.001; T = trivial effect size; S = small effect size; M = moderate effect size; L = large effect size.

**Table 4 ijerph-17-08272-t004:** Binary logistic regression analysis with postponed or cancelled routine health care appointments (yes vs. no) as dependent variable, and illness severity, health anxiety and stress as predictors.

Dimension	Variables	Coefficient	Standard Error	Wald	*p*	Nagelkerke R^2^
Postponing/cancelling appointments (yes vs. no)	Constant	−18.97	5.64	11.31	0.002	0.85
	Illness severity	0.507	0.206	6.05	0.014	
	Health anxiety; total	0.496	0.147	11.30	0.001	
	Stress	−0.161	0.068	5.66	0.017	
Excluded variables	Illness likelihood, body vigilance, depression (all Wald’s < 1.8, all *p*’s > 0.10.					

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
