# Peer review of "Health Anxiety Predicts Postponing or Cancelling Routine Medical Health Care Appointments among Women in Perinatal Stage during the Covid-19 Lockdown"

_ijerph, 2020, doi:10.3390/ijerph17218272_

Round 1
Reviewer 1 Report
First, I would like to thank for the opportunity of reviewing this manuscript entitled "Health anxiety predicts postponing or cancelling routine medical
health care appointments among women in perinatal stage during the COVID-19
breakdown" which discusses the psychological responses to the COVID-19 pandemic of a special population represented by females in the prenatal or postnatal stage. Additionally, the approach to the healthcare system is evaluated. Despite the topic is of extreme interest, several papers have been already published, and the small sample size of the present study does not prioritize its acceptance.
Major comments:
- The introduction looks too long and in my opinion it may distract from the key message of the study. I suggest shortening and eventually transferring some paragraphs to the discussion.
- Compared to other studies on the same topic, this study seems to be undersize and therefore results should be taken with greater caution. As such, I recommend to include i) a power analysis, or ii) to be more cautious in claiming the results in light of the small sample size.
- Tables layout does not help its reading and understanding, and it makes difficult to me to follow the findings. I suggest adapting columns width to make the reading more linear. You could also consider to use some chart figures.
- To me, the novelty comes from the association between the psychological scores and the decision of postponing/cancelling routine medical care appointments during these hard times of the pandemic; the results are too superficially discussed in the results section and a greater description of the model should be expanded (e.g., I guess a multivariate binary logist regression was run; how factors were included in the multivariate analysis? Were they previously tested with an univariate analysis and only those with a sufficient significance were then included). If you well described it in the tables, I am sorry but it was really hard to me to read it.
Minor comments:
- Pag 3 Lines 127: Can you please specify how participants were recruited to the study?
- Pag 3 Lines 140: As I can understand, participants had to be in the prenatal or postnatal stage no longer than six weeks ago (do you mean, < 6 wk prior the participation to the survey?). Can you further describe the sample in the terms of what stage of the pregnancy/postpartum correspond the included stages? E.g., the prenatal stage corresponded to what range of the pregnancy? The same for the post-partum, how many weeks after delivery it corresponds?
- Discussion: I understand that maybe, at the time of submission, this was the only paper "to investigate the 264 impact of the COVID-19 pandemic among females in a perinatal stage". Unfortunataly, at the time of the review, this is not totally true. Different aspects, including psychological evaluation, have been already described by other authors also in larger samples (e.g., Lebel et al., 2020; Berthelot et al., 2020; Masjoudi et al., 2020; Barisic, 2020; Moyer et al., 2020; etc.), as well some data on access to the healthcare service (Monni et al., 2020; Ahlers-Schmidt et al., 2020; Wang et al. 2020, Justman et al., 2020; etc.). I suggest removing that sentence.
- Discussion: I suggest adding a paragraph about possible interventions to help those mothers to reduce the psychological impact of the pandemic and to promote the access to the healthcare and medical appointments, as for example psychological counseling and the development of telehealth protocols (Jeganathan et al., 2020).
Author Response
Dear Reviewer, thank you for all your kind efforts. Please see the detailed point-by-point-response.

Reviewer 2 Report
The aim of the present study was to investigate the associations between the subjective belief of risk for infections and health anxiety, depression, stress and further perinatal dimensions. The main finding was that health anxiety was related to postponing or canceling routine medical check appointments. However, health anxiety was not associated with stress and depression.
The study covers an interesting and actual topic that, considering the situation related to the COVID-19, is worth being investigated. The manuscript is well-written and easy to read. The introduction is clear and provides an exhaustive overview of the literature related to the topic. Methods and results are well-described and presented. Discussion is complete and precise, elaborating sequentially each of the three research questions. The conclusion section contextualizes the main findings in the light of practical and clinical perspectives. Overall, I would like to congratulate the Authors for their work.
I have only a point to suggest to discuss. The relationship of anxiety, stress, depression with physical activity is well known in the literature, suggesting that physical activity may have a role in reducing such cognitive-emotional dysfunctions. Did the Authors assess also the physical activity levels of participants? Please elaborate. If not, I suggest to include this issue in the limitations of the study.
Author Response
Dear Reviewer, thank you for all your kind efforts; please see the detailed point-by-point-response.

Reviewer 3 Report
The Shayganfard et al manuscript show how the COVID-19 world pandemic impact the mental health in the pregnant and breastfeeding women. The lockdown has been and continuous being an unexpected situation for all countries. Their health impact is unpredictable and this work are very interesting to know how affects into the mental health and how the professional health cares need to take our populations. However, I would kindly suggest some details and comments which could improve the understandable of the text.
Title: I suggest the title been changed for something more focus such as “Health anxiety predicts postponing routine health care appointment among women in perinatal stage during COVID-19”
Abstract: Review the journal guidelines, the paragraph should not have subheadings. In addition, the age of the cohort was reported in the results section and in the abstract was included in methods. The last sentences of the abstract do not match with the rest of the format.
Introduction: in general, the section should be more summarized and focused. Particularly:
- Why is relevant the Multiple Sclerosis paragraph in the context of the manuscript?
- The paragraph between lines 73 to line 89 need to be strongly summarized because the dilute the flow of the work.
- The health anxiety concepts should be re-written, its difficult to understand what the authors want to say. In general, between lines 94 to 99, there are multiple concepts which need to define more accurate, maybe, could be synthetized/summarized?
- Sentence in line 103-104 is redundant and could be deleted.
- Lines 108 to 119 should be summarized.
- The concepts under paragraph between line 120 to line 124 are repeated in the rest of the section. I suggest remove the lines.
Method: In the psychological applications, the authors should report if the consistency was their self or reported by others and more psychometric properties, as well as a Likert scale. In the application of behavior related to the COVID-19 if was ad-hoc and there was a pitot study to report the reliability.
- Section 2.2. should be “enrollment protocol” or similar. And section 2.3. should be “applications”
- What was the language used? there was any exclusion/inclusion criteria regarding to? How many time the women used to response the survey? In addition, the authors could report some example which they considered severe psychiatric issue and severe psychological issue, including surgery.
- Line 147, what test was used? RT-PCR, serology detection, by capillarity test, etc. Please specify. They need to report the company used as well as a city and country of the company.
- In the sociodemographic information, was controlled by abortion/miscarriage? Could be a modulatory factor in the pregnancy anxiety.
- In the health anxiety applications, the diagnoses of hypochondriasis, would not be tested in the exclusion criteria?
- The stat. section need to be strongly re-organized describing as:
- What descriptive measurement were used and according what criteria (maybe normality of the variables?)
- What univariate analysis was applied: what mean-test, what correlation test, how define the ANOVA (singular)
- What multivariate or model analysis was applied, how the modulatory or cofounders were treated.
- What p-value was considered as a significant.
- What stat software, version and company was used.
Results: The tables should have some details of the stat. in their footnotes. The table 3 is difficult to understand and did not match with the journal guidelines format. I suggest re-structure it, maybe without stat. scores.
- Subheading 3.1. is not necessary.
- Please, report the descriptive as 28.6±7 years old (line 212)
- Line 217 is not necessary.
- The statistic column in table is not necessary, more adequacy would be report the p-value or confidence intervals. The “entire sample” should say “Total”. The gestational age was 17.2±8, “before delivery”, this group could re-name as “pregnancy”. In addition, in the table, the GA variable should specify “complete”
- Line 229, trivial to low would be subjective attributes which should lead the discussion section not results.
- Table 2. The authors should name the variables as a scale names.
- In the section 3.4. in the beta coefficients should be added the standard error. In addition, the models could have more clinical relevant if the authors would report the OR and 95% confidence intervals.
Discussion: In general, is a well-structured section, the authors clear the three research-questions and hypothesis which link with their results in the manuscript.
Conclusion: I strongly suggest an abstract figure with the Health Anxiety as a core-concept and how this is related to “postpone the health care appointment” and how this concept is unrelated with depression and stress.
- Sentence in the line 342-343, I suggest would be treated with caution.
Minor comments:
- I recommend change the expression of “female” for “women” and unification the words as “participants” (i.e.: line 57)
- Lines 133 and 135, “see below” could be delated.
- The variables “yes vs no”, please, mention it as a binary or dichotomies variable.
- There is some type error, please double-check it (i.e. “)” missing, line 220)
Author Response
Dear Reviewer,
Thank you for all your kind efforts.
Please see the attached point-by-point-response.

Round 2
Reviewer 1 Report
I have to thank and congratulate the authors for the prompt and successful revision of my comments, and in my opinion the manuscript really improved. I have no further comments.
Author Response
Thank you again for the care you have devoted to review the manuscript. Please find the detailed point-by-point-response attached as a separate file.

Reviewer 3 Report
The Shayganfard et al manuscript has been extremely re-written, the methodology is clearer and focus then previous version. In addition, the format of the table is now readable. I have some doubts with the introduction and results section, which maybe could be decide for the editorial officer.
Introduction: Although it is appreciable the re-written in the text, I think this section is extremely large and could dilute the main focus of the article. Continues redundant sentences and no-focus paragraphs.
- Line 102. COVID-19 means the disease, not exactly the virus.
Results: Although previously the report in stats was good feeling, if nothing say related to the distribution of the variable the stat-score is not informative (more than which p-value report). What could I compare with χ2=0.19 whit 2 degrees of freedom and 103 observation if I do not have the contrast distribution? However, with the random probability and (in case) the relative risk of this proportion or the confidence interval I can determinate if this variable had a random-probability and also it was associated or not.
- I continue thinking the Subheading 3.1. is not necessary.
Minor comments:
- I did not see the figure 1.
Author Response
Thank you again for the care you have devoted to review the manuscript. Please find the detailed point-by-point-response attached as a separate file.

This manuscript is a resubmission of an earlier submission. The following is a list of the peer review reports and author responses from that submission.